

# The salivary gland salivation stimulating peptide from *Locusta migratoria* (Lom-SG-SASP) is not a typical neuropeptide

Jan A. Veenstra

CNRS, Institut de Neurosciences Cognitives et Intégratives d'Aquitaine (UMR5287), Université de Bordeaux, Pessac, France

## ABSTRACT

The salivary gland salivation stimulating peptide was identified from the salivary glands of the migratory locust by its ability to stimulate cAMP production in the same tissue. The gene coding for this peptide has recently been identified and been shown to code for a precursor consisting of a signal peptide, several copies of the peptide separated by Lys–Arg doublets and a few other peptides. These data are consistent with it being a neuropeptide. However, antiserum raised to this peptide labels the acini of the salivary glands while RT-PCR only gives positive results in the salivary gland, but not in any ganglion of the central nervous system. Thus, this peptide is not a typical neuropeptide as previously assumed.

## INTRODUCTION

Insect neuropeptides are interesting for two different reasons. On the one hand, insects are protostomes, while vertebrates are deuterostomes. Thus, comparing insect with vertebrate neuropeptide regulatory systems provides insight as to how structures and functions may have changed during evolution. On the other hand, many insect species are pests and vectors of disease and thus responsible for serious agricultural damage as well as the transmission of human disease. As insects are becoming increasingly resistant to classical pesticides, novel insecticides are constantly needed and it is has repeatedly been suggested that agonists or antagonists of insect neuropeptide receptors might offer a solution (*Audsley & Down, 2015*; *Van Hiel et al., 2010*). Inhibition or at least disruption of feeding by insects would be very attractive as it would presumably avoid or diminish economic damage or, in the case of disease vectors, might reduce transmission of infectious agents. As production of saliva is usually a first and necessary step in feeding the regulation of salivation by neuropeptides is particularly interesting.

A pentadecapeptide was isolated from the salivary glands of the migratory locust by its ability to stimulate the production of cAMP in the same glands at concentration of $10^{-6}$ M (*Veelaert et al., 1995*). As it also stimulates salivation (*Veelaert et al., 1995*), its characteristics suggests it to be neuropeptide that likely acts as a neurotransmitter rather

Corresponding author
Jan A. Veenstra,
jan.veenstra@u-bordeaux.fr

MALKTLAAFLLVVCLAQLTCASPAPAPNRRSTDESTADSSSATVTTSANNATS
DADTISSNDTSSDDIFGEIGEKVEGFFGHLFGKRDIEIPSDVFTKLYQEWAKG
RPSRSVTVRDTGKNFTVGDLFQEWLQGNVNKRSVTVREVGNLFQEWLQGNVNK
RSVTVREVGDLFQEWLQGNVNKRSVTVREVGDLFQEWLQGNVNKRSVTVREVG
DLFQEWLQGNVNKRSVTVREVGDLFQEWLQGNVNKRSVTVREVGDLFQEWLQG
NVNKRSVTVREVGDLFQEWLQGNVNKRSVTVREVGDLFQEWLQGNVNKRSVTV
REVGDLFQEWLQGNVNKRSVTVREVGDLFEEWLQGNMNKRSVTVRDTDNSSTM
GDLIKEWLKGNLNERSVTVRDTGKNFTVGELFQEWLQGNVNKRSVTVRDTDNS
STMNERSVTV

**Figure 1 Conceptual translation of the predicted open reading frame of SG-SASP from *Locusta migratoria* conceptual translation of the SASP mRNA predicted from transcriptome and genome sequences (*Veenstra, 2016*).** Amino acid sequences highlighted in yellow indicate the signal peptide, Lys–Arg doublets highlighted in red are predicted convertase cleavage sites. In blue are the multiple copies of SVTVREVGDLFQEWLQGNVN, the major peptide encoded by the SASP gene. The amino acid sequence obtained by *Veelaert et al. (1995)* is not present in this sequence, but there is a single copy of EVGDLFEEWLQGNMN that is highlighted in black. The molecular mass of the EVGDLFEEWLQGNMN differs by less than 1 Da from that of EVGDLFKEWLQGNMN. It seems plausible that an incorrect interpretation of the Edman degradation is responsible for the difference between the peptide sequence as experimentally determined and that deduced from the various DNA sequences. Similar minor differences were found for other *Locusta* neuropeptides (*Veenstra, 2014*). EVGDLFEEWLQGNMN is predicted to be cleaved from its precursor by typical neuropeptide convertases. The Lys–Arg doublet at its C-terminal in the precursor is the canonical neuropeptide precursor cleavage site, while cleavage at the single Arg residue at its N-terminal is supported by the presence of an Arg residue at the −6 position (*Veenstra, 2000*). The latter Arg residue is part of another Lys–Arg doublet, that is the preferred convertase substrate and should thus be expected to be cleaved rapidly on exposure of the precursor to convertase. Once this site has been cleaved, the single Arg residue can no longer be cleaved. Consequently, it must be expected that little of EVGDLFEEWLQGNMN will be produced as most of the time the Lys–Arg site will be processed preferentially and once this has occurred there is no longer an Arg residue in position −6 to support cleavage at the single Arg residue. Based on the precursor sequence and known convertase cleavage preferences in insects (*Veenstra, 2000*) one would expect it be processed mostly into SVTVREVGDLFQEWLQGNVN, the peptide used for raising an antiserum.

than a hormone. If it were a hormone, one would expect it to stimulate the production of cAMP and salivation in the nanomolar, rather than in the micromolar range and it would be expected in a neurohemal organ, rather than in the salivary gland itself (*Veelaert et al., 1995*). Unlike most insect neuropeptides orthologs of this peptide have not been identified from any other arthropod, suggesting that it may not be universally present in insects. This might be advantageous as any pesticide based on it could be relatively selective. The genome sequence combined with the RNAseq data of the migratory locust (*Wang et al., 2014*) made it possible to predict a likely precursor for this peptide (*Veenstra, 2014*). There is a single amino acid mismatch between the predicted precursor and the identified peptide, but the peptide sequence is bounded by putative convertase cleavage sites (Fig. 1) and it is thus highly likely that this is indeed the precursor of this peptide. The precursor has all the hallmarks of a typical neuropeptide precursor: a signal peptide and a propeptide encoding multiple copies of the peptide separated by Lys–Arg convertase cleavage sites. It thus appeared of interest to study this putative neuropeptide in more detail, but as reported here the peptide turned out not to be a typical neuropeptide, in spite of previous data suggesting otherwise.

## MATERIALS AND METHODS

### Locusts

Adult and fifth instar *Locusta migratoria* were purchased at a local pet store. They were kept for five to seven days at 25 °C and fed fresh grass once a day before being used. Tissues were dissected under saline and either frozen immediately at −80 °C for subsequent RNA extraction, or used for immunohistology.

### Immunohistology

I chose the SVTVREVGDLFQEWLQGNVN sequence for making antisera, as this sequence is present in multiple copies on the precursor (Fig. 1). Two milligrams (purity 84%, Proteogenix, Schiltigheim, France) were conjugated to 5 mg of bovine serum albumin using difluorodinitrobenzeze as the conjugation reagent as documented by *Tager (1976)*. Polyclonal mouse antisera were raised in three 6-week-old NMRI female mice as described previously (*Veenstra & Ida, 2014*). Tissues were fixed for 1–2 h at room temperature. All other immunohistological procedures are the same as described (*Veenstra & Ida, 2014*). Primary antiserum was diluted 1:2,000, the secondary antiserum, DyLight-488-conjugated goat anti-mouse IgG that was from Jackson ImmunoResearch Europe (Newmarket, Suffolk, UK), 1:1,000.

### Bioinformatics

cDNA sequences coding for the *L. migratoria* orthologs of vertebrate PC1 and PC2 are not present in the databases and were therefore obtained by using a combination of the published genome sequence and Trinity on sequences extracted from the various short read archives (SRAs) for this species available at NCBI (SRR014351, SRR014352, SRR058432, SRR058446, SRR058447, SRR058448, SRR058449, SRR058450, SRR058451, SRR058452, SRR058453, SRR058454, SRR058455, SRR058456, SRR058457, SRR058458, SRR058488, SRR058489, SRR058490, SRR058491, SRR058492, SRR058493, SRR058494, SRR058495, SRR058496, SRR058497, SRR058498, SRR058499, SRR058500, SRR058501, SRR058502, SRR058503, SRR1032161, SRR1032192, SRR167712, SRR513208, SRR513209, SRR513210, and SRR513211) using methodology described in detail elsewhere (*Veenstra, 2016*). Protein and cDNA sequences for *L. migratoria* PC1 and PC2 are provided in the Data S1.

### RT-PCR

The following tissues were dissected: brain, suboesophageal ganglion, pro- and meso-thoracic ganglia combined, the meta-thoracic ganglia combined with all abdominal ganglia, salivary gland, fat body, and midgut. For the analysis of the expression of PC1 and PC2 Malpighian tubules and flight muscle were also analyzed. At least two samples were processed completely independently for each of these tissues. Each sample containing tissues from at least four different animals. From salivary glands, fat body, Malpighian tubules, and flight muscle, small parts were taken from four different animals. For the midgut four longitudinal half midguts were processed individually. RNA extraction was performed using mini spin columns from Macherey-Nagel. Next RNA

(800–1,000 ng) was reverse transcribed in a 20 µl reaction using Moloney murine leukemia virus reverse transcriptase (New England Biolabs, Evry, France) and random primers. One microliter of the resulting cDNA was next amplified by PCR using OneTaq Quick-Load DNA Polymerase (New England Biolabs) with specific primers for each mRNA. Primers used are: for the salivary gland salivation stimulating peptide: 5′-GCCTTCCTGCTAGTCGTCTG-3′ and 5′-TACCTTTTGCCCACTCTTGG-3′; for actin: 5′-AGGGCTGTTTTTCCCTCAAT-3′ and 5′-GAAGGTGTGGTGCCAGATTT-3′; for PC1: 5′-ACAACCACGTGCACAAGAAG-3′ and 5′-TGAATGCGACTAAGCCACAG-3′; and for PC2: 5′-GGTGGACTACCTGGAACACG-3′ and 5′-TGTGGATATTCTCCCCAGGT-3′. PCR profiles consisted of 90 s denaturing at 94 °C followed by 32 cycles of 30 s at 94 °C, 15 s at the annealing temperature and 15 s at 68 °C the amplification was followed by 5 min extension at 68 °C. Annealing temperatures were 55, 64, 64, and 60 °C for the salivary gland salivation stimulating peptide, actin, PC1, and PC2, respectively. Controls in which water replaced the cDNA showed no PCR amplification.

## RESULTS AND DISCUSSION

I anticipated that an antiserum would allow the identification of neurons expressing this peptide. However all three mice produced antisera that labeled the acinar cells of the salivary glands (Fig. 2), while immunoreactive material was completely absent from the central nervous system. Pre-immune antisera from the same mice or diluted antisera that were absorbed with SVTVREVGDLFQEWLQGNVN (overnight, 10 µg/ml) were not immunoreactive (Fig. 3), thus showing that the observed immunoreactivity is specific. This suggested that the peptide was made by the salivary gland itself, rather than by the nervous system. Intron spanning primers were designed to look for expression of the gene coding the peptide and results revealed amplification only in the salivary glands (Fig. 4). The PCR-amplified band was sequenced using the primers for amplification and the sequencing results confirmed the expected sequence (Fig. S1). Thus there is independent confirmation that this gene is expressed in the salivary gland, but neither in the central nervous system, the fat body nor the midgut. RT-PCR confirmed strong expression of this gene as (1) amplicons become visible after 20 cycles (Fig. 4) and (2) the slightest contamination of thoracic ganglia with a small piece of salivary gland (these tissues are closely associated with one another) leads to false positives. The integrity of the cDNA samples were checked by looking for expression of actin as a control.

The Lys–Arg convertase cleavage sites in the precursor are identical to those commonly found in neuropeptide precursors and thus suggested the presence of a neuropeptide specific convertase in the salivary gland. The cDNA sequences of the two locust orthologs of vertebrate PC1 and PC2 were obtained using Trinity on the various SRAs for this species available at NCBI (Fig. S2) and intron spanning primers were designed to look for their expression in the salivary gland. PC1 was found not only in the salivary gland, but also in other peripheral tissues (Fig. 5), but PC2 was absent from the salivary glands. It thus appears that the tissue distribution of PC1 in migratory locusts and perhaps other arthropods is much broader than in vertebrates (Seidah et al., 2013).

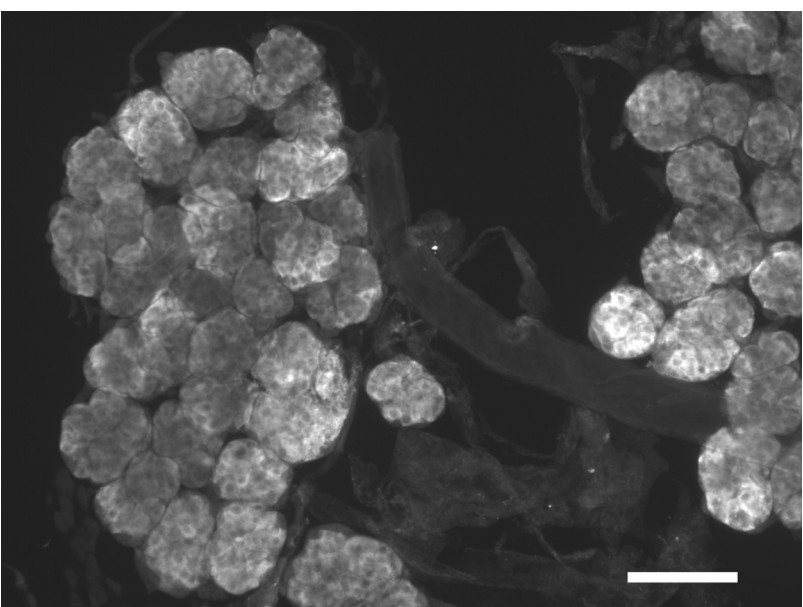

**Figure 2** **Immunohistological localization of SG-SASP.** Immunoreactive salivary gland peptide in the adult salivary gland of *L. migratoria*. Note the strong labeling in the acinar cells. Scale bar 250 μm.

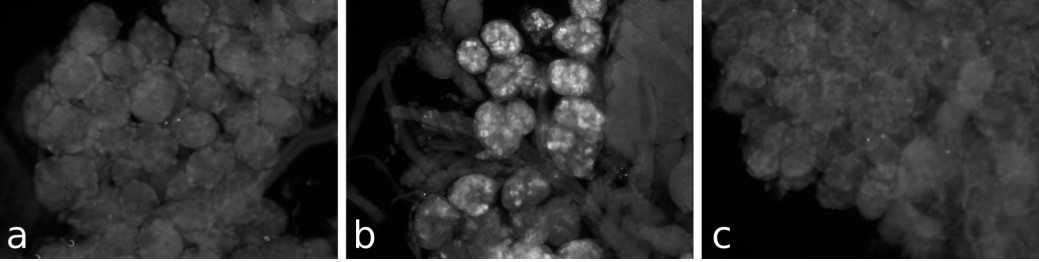

**Figure 3** **Antiserum specificity.** Different pieces of the same salivary gland were exposed to pre-immune serum (A), immune serum (B), or immune serum pre-absorbed with SVTVREVGDLFQEWLQGNVN (C). Note that no immunoreactivity is found in the pre-immune serum and the immunoreactivity is abolished after pre-absorption to SG-SASP.

A question that remains to be answered is what the function of this peptide might be. Salivary glands typically secrete digestive enzymes in an exocrine fashion, but in principle, it could also be secreted on the basolateral membrane into the hemolymph. Only the latter possibility could explain how it could stimulate the salivary glands to produce cyclic AMP. In spite of this and the precedent for autocrine stimulation by dopamine in the tick salivary gland (*Koči, Šimo & Park, 2014*), there are several arguments to suggests that the peptide is secreted as an exocrine rather than an autocrine or paracrine. First, the situation in ticks remains highly unusual. Secondly, in the tick dopamine is made only in a minor cell type, while SASP is made in the major cell type. Thirdly, SASP is made as a protein precursor and as such should be expected to be co-secreted with the digestive enzymes in the same granules (this is not the case for dopamine, which is present in different granules). It would be very surprising if such

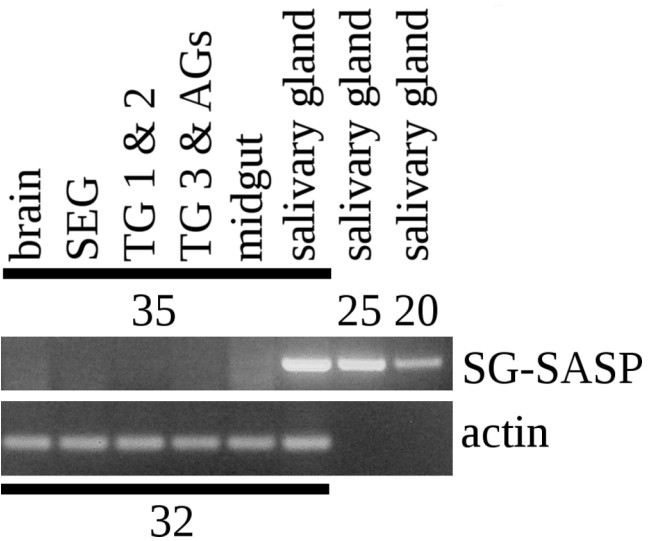

**Figure 4 RT-PCR of the salivary gland peptide.** RT-PCR on different tissues from *L. migratoria* for the expression of SG-SASP and actin. Numbers indicate the number of PCR cycles employed. Note that 20 cycles is sufficient to show expression in the salivary gland, while 35 cycles do not show any expression in the nervous system. TG 1 & 2, the pro- and meso-thoracic ganglion combined, TG 3 & AGs the meta-thoracic ganglion together with all the abdominal ganglia. Complete gels with markers are shown in the Data S1.

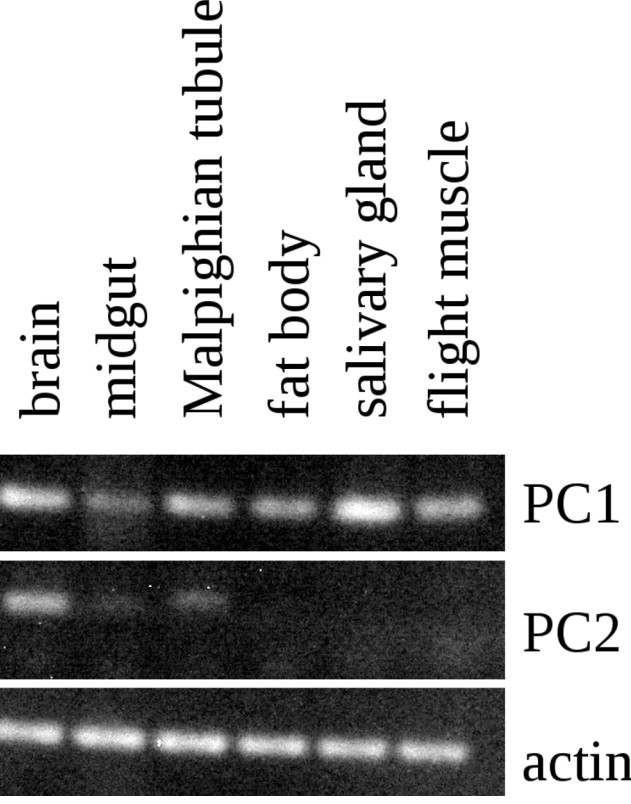

**Figure 5 RT-PCR of PC1 and PC2.** RT-PCR on different tissues from *L. migratoria* for the expression of PC1 and PC2 convertases with actin as a control. Note that PC1 is widely expressed, including in the salivary gland, while the expression of PC2 is much more limited and not expressed in the salivary gland. Thirty-two cycles of PCR in each sample. Complete gels with markers are shown in Data S1.

digestive enzyme containing granules would be secreted into the hemolymph. Fourth, although dopamine in the tick salivary gland is present in relatively high concentrations, it remains a minor product, while SASP as suggested by the immunohistological data as well as the RT-PCR seems to be a major product of the salivary gland. Fifth, it is to be expected that the Lys–Arg cleavage sites in the SASP precursor are cleaved efficiently; however, for the peptide identified by *Veelaert et al. (1995)* to be produced, this site needs to be still there, for a single Arg residue is not easily cleaved by convertase. Hence, relatively small amounts of the active peptide are expected to be made, while the likely much larger amounts of the peptide used here as antigen, were not identified as a stimulator of cyclic AMP production. This seems to suggests that the stimulation of cyclic AMP production is an accidental artifact, as perhaps also suggested by the high concentrations needed of this peptide to do so.

Even if this may suggest an exocrine function for this peptide, it does not answer the question what this function entails. As saliva is generally reabsorbed with the food, its tempting to speculate that it somehow facilitates digestion.

In the last decade a large number of putative neuropeptide genes have been identified in genome sequences of a wide variety of invertebrate species, sometimes based exclusively on the presence of signal peptide and Lys–Arg cleavage sites that separate the presumptive neuropeptides in the precursor. In many cases the predicted peptides show clear homology to known neuropeptides, while in other cases they have subsequently been shown to activate G-protein coupled receptors (*Bauknecht & Jékely, 2015*). However, as illustrated here the presence of a signal peptide, reputable convertase cleavages sites and multiple copies of the same or a very similar peptide do not make a neuropeptide precursor.

### Funding
This work was supported by institutional funding from the CNRS. The funders had no role in study design, data collection and analysis, decision to publish, or preparation of the manuscript.

### Grant Disclosures
The following grant information was disclosed by the authors:
CNRS.

### Competing Interests
The author declares that he has no competing interests.

### Author Contributions
- Jan A. Veenstra conceived and designed the experiments, performed the experiments, analyzed the data, contributed reagents/materials/analysis tools, wrote the paper, prepared figures and/or tables, reviewed drafts of the paper.

## Data Availability

The raw data has been provided as Supplemental Dataset Files.

## Supplemental Information

Supplemental information for this article can be found online at http://dx.doi.org/10.7717/peerj.3619#supplemental-information.

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
