# Peer review of "The salivary gland salivation stimulating peptide from Locusta migratoria (Lom-SG-SASP) is not a typical neuropeptide"

_PeerJ, doi:10.7717/peerj.3619_

## Round 0.1 · original submission · Minor Revisions

Both reviewers agree that the paper in its present form lacks a detailed descriptions of SASP, especially as it seems that the sequence is from Veelaert et al (1995). One of the reviewers actually suggests an additional figure which I find a very good idea, as this could be nicely combined with a description/explanation in the ms text itself. Apart from this, there a couple of additional issues risen ("neuropeptide", Antibody characterization etc) by the reviewers that should be carefully addressed in a revised version of the manuscript. The required revisions are somewhat on the border between "minor" and
major" but I finally decided to stick to "minor" revisions to encourage a fast resubmission.

Reviewer 1 ·

Basic reporting

The manuscript describes an investigation of salivation stimulating peptide (SASP) of Locust migratoria. Interestingly, the author concluded that the SASP is NOT a neuropeptide because it is produced from the salivary gland itself.

Overall, I believe that the major finding of the manuscript, SASP produced in the salivary gland acini, is a significant knowledge to be published. Although the functions of SASP discussed above is an important issue, the expanded work will hold up publication of this important finding. Therefore I do not insist the author include functional study in this manuscript. However, I strongly suggest revision for following three aspects;
1. Rephrasing the major message of “not a neuropeptide” included in the title
2. The gene structure, cDNA sequence, and the putative translation.
3. Antibody characterization

Experimental design

In order to conclude “not a neuropeptide”, authors need to define “neuropeptide” first. I am not sure how we define “neuropeptide” in this case. I make definition of “signaling peptide” as any small peptide that functions as an endocrine hormone/neural factor (neuromodulator and neurotransmitter). Classical definition of “neuropeptide” may be signaling peptides produced and secreted from neural organ (or cells). Unfortunately, majority of neuropeptides are now found to be produced and secreted from both in neural and non-neural cells as the author has previously published for many neuropeptides produced in enteroendocrine cells (and other organs). Therefore, the title and the main message saying SASP is NOT a neuropeptide is not appropriate issue to deal with in this manuscript because of confounding problem in the definition of neuropeptide in insect world. I understand that the “not a neuropeptide” meant “not a neuropeptide in innervation of salivary gland”. Therefore, I suggest author rephrases the title and other related sentences.

In addition, if the SASP functions as an autocrine-endocrine factor could be a very likely the case combining the knowledge from earlier work of Veelaert et al (1995) and from this study. Autocrine function of dopamine in tick salivary gland was previously shown (Koci et al., 2014, J. Ins. Phys.), implying a similar mechanism may occur for the locust SASP. In the case of tick, the autocrine factor dopamine may activate the salivary glands and other organ such as cybarial pump, implying that the SASP may activate expanded repertoire of functions in addition to the activation of salivary gland acini.

Alternatively, based on the SASP produced in the acini of salivary glands, an immediate question is whether it is an exocrine factor, and the question is further expanded whether the exocrine factor secreted and orally re-uptaken SASP can function on the gut and other internal organ. This hypothesis can be tested by having a good tool, specific antibody as author has developed.

Validity of the findings

The details of the descriptions of SASP is lacking in the manuscript. Fist of all, the sequence SVTVREVGDLFQEWLQQNVN is different from the sequence identified in Veelaert et al (1995) as EVGDLFKEWLQGNMN. The first five AAs SVTVR was not found in Veelaert and three additional internal mismatches as Q/K, Q/G, V/M are found. Author need to describe the gene structure, cDNA sequence and putative translation and explain why this sequence is used for epitope in raising the antibody. If it is available, details of the descriptions on the cDNA structure and the signal peptide could be added.

The author needs to characterize the antibody by showing preimmmune serum (which might be unavailable in the case of mouse sera) or pre adsorbtion of the antisera.

Reviewer 2 ·

Basic reporting

Suggested improvements:
- Check lines 117-118 for a spelling and word-run on errors
- A figure with the amino acid sequence of the predicted open reading frame of the Lom-SG-SASP with the antigen region and motifs highlighted that are discussed.

Experimental design

No comment

Validity of the findings

Suggested improvement:
Since the SG-SASP is produced by gland cells and presumably stored or secreted into the gland duct, the author should provide a bit of speculation as to how the SG-SASP could stimulate salivation and cAMP by the glands as shown by the founding paper (Veelaert et al., 1995). It may be possible that the peptide is released into the hemolymph where it could have autocrine effects on the glands, but this is not the usual mode for salivary gland cells.

---

## Round 0.2 · accepted · Accept

The author provided a very careful revision of the initial manuscript and I feel that all issues raised by the reviewers and me were sufficiently addressed or discussed. Therefore I consider the manuscript now as ready for publication.